# Isolation, Characterization, and Safety Evaluation of the Novel Probiotic Strain *Lacticaseibacillus paracasei* IDCC 3401 via Genomic and Phenotypic Approaches

**DOI:** 10.3390/microorganisms12010085

**Published:** 2023-12-31

**Authors:** Han Bin Lee, Won Yeong Bang, Gyu Ri Shin, Hyeon Ji Jeon, Young Hoon Jung, Jungwoo Yang

**Affiliations:** 1Ildong Bioscience, Pyeongtaek-si 17957, Republic of Korea; gksqls9131@ildong.com (H.B.L.); yeong0417@ildong.com (W.Y.B.); 2School of Food Science and Biotechnology, Kyungpook National University, Daegu 41566, Republic of Korea; li824@naver.com (G.R.S.); hhyyeeoonnji@gmail.com (H.J.J.)

**Keywords:** probiotics, safety assessment, *Lacticaseibacillus paracasei*, genome sequencing, lactic acid bacteria

## Abstract

This study aimed to explore the safety and properties of *Lacticaseibacillus paracasei* IDCC 3401 as a novel probiotic strain via genomic and phenotypic analyses. In whole-genome sequencing, the genes associated with antibiotic resistance and virulence were not detected in this strain. The minimum inhibitory concentration test revealed that *L. paracasei* IDCC 3401 was susceptible to all the antibiotics tested, except for kanamycin. Furthermore, the strain did not produce toxigenic compounds, such as biogenic amines and D-lactate, nor did it exhibit significant toxicity in a single-dose acute oral toxicity test in rats. Phenotypic characterization of carbohydrate utilization and enzymatic activities indicated that *L. paracasei* IDCC 3401 can utilize various nutrients, allowing it to grow in deficient conditions and produce health-promoting metabolites. The presence of *L. paracasei* IDCC 3401 supernatants significantly inhibited the growth of enteric pathogens (*p* < 0.05). In addition, the adhesion ability of *L. paracasei* IDCC 3401 to intestinal epithelial cells was found to be as superior as that of *Lacticaseibacillus rhamnosus* GG. These results suggest that *L. paracasei* IDCC 3401 is safe for consumption and provides health benefits to the host.

## 1. Introduction

The commercial use of lactic acid bacteria (LAB) in the food industry over the recent decades [1] led to the isolation and study of various LAB, including *Lacticaseibacillus, Lactococcus*, and *Bifidobacterium*, to improve the quality of human life. LAB have demonstrated their potential role as probiotics, promoting health and inhibiting the growth of pathogenic bacteria [2]. Probiotics are live microorganisms that, when administered in adequate amounts, bring about a positive change in the gut microbiota of the host through competitive growth with pathogens and the production of beneficial substances [3]. Previous studies have investigated their diverse functions, including their ability to modulate the immune system, exert anticancer and antiobesity effects, and reduce cholesterol [4,5,6].

Although probiotics have been shown to provide health benefits, the use of newly isolated microorganisms in humans has been associated with undesirable effects such as resistance against antibiotics and the production of toxigenic compounds [7]. Therefore, it is important to verify their safety by testing their toxicity or infectivity before being used in humans [8]. The safety of a novel probiotic strain must be evaluated using in vivo and in vitro models according to international guidelines [8], including those of the Food and Agriculture Organization/World Health Organization (FAO/WHO) and the European Food Safety Authority (EFSA) [3,9].

Probiotics exert beneficial effects on the host by maintaining sufficient viable cells and unique properties during transit through the stomach and small intestines before reaching the large intestines [10]. These functionalities depend on their ability to resist the stressful environment and to compete with the growth of pathogenic bacteria and settle on the intestinal tract [9]. Hence, the selection criteria of probiotic strains should include carbohydrate utilization diversity, enzymatic activities, acid and bile tolerance, antimicrobial activities, and adhesion to intestinal epithelial cell lines [11].

*Lacticaseibacillus paracasei* is one of the most commonly used bacteria for food fermentation and was first proposed as a novel probiotic species in 1989 [12]. Its health benefits include potential antimicrobial, anti-inflammatory, and stress-modulating effects [13]. However, using probiotics without fully establishing their safety has been reported to cause diarrhea, gastrointestinal ischemia, and abdominal pain [14]. In addition, not all bacteria of a given genus or species have probiotic properties [15]. Therefore, although *Lacticaseibacillus* are commonly found in vegetables, dairy products, and meat and have long been used in the food industry [16], the safety of *L. paracasei* IDCC 3401 for use as a novel probiotic strain and its properties need to be investigated further.

The present study examined the safety and properties of *L. paracasei* IDCC 3401 as a potential probiotic via phenotypic and genomic analyses, including genome sequencing, minimum inhibitory concentration (MIC) test, biogenic amine (BA) production analysis, L-/D-lactate production analysis, single-dose acute oral toxicity test, acid and bile tolerance test, carbohydrate utilization test, enzymatic activity assay, antipathogenic activity, and adhesion assay.

## 2. Materials and Methods

### 2.1. Bacteria Culture Conditions

*L. paracasei* IDCC 3401 (ATCC No. BAA-2839) was isolated from the feces of breast-milk-fed infants by Ildong Bioscience (Pyeongtaek-si, Korea). This strain was anaerobically grown in de Man, Rogosa, and Sharpe (MRS) agar (BD Difco, Franklin Lakes, NJ, USA) at 37 °C. Cell growth was measured at 600 nm using a microplate spectrophotometer (EPOCH2, BioTek, Winooski, VT, USA). The four pathogenic microorganisms used in the experiments, namely, *Staphylococcus aureus* ATCC 25923, *Enterococcus faecalis* ATCC 29212, *Bacillus cereus* ATCC 14579, and *Salmonella* Typhimurium ATCC 13311, were purchased from the American Type Culture Collection (ATCC). The pathogens were incubated at each culture condition listed in Appendix A.

### 2.2. Genomic Analysis of Lacticaseibacillus paracasei IDCC 3401

The genomic DNA of *L. paracasei* IDCC 3401 was extracted using a Wizard Genomic DNA Purification Kit (Promega Co., Madison, WI, USA) according to the manufacturer’s instructions. The genome of strain IDCC 3401 underwent comprehensive sequencing utilizing both Pacific Biosciences (PacBio, Menlo Park, CA, USA) RS II and Illumina iSeq 100 systems. PacBio RS II data were processed using the Canu v1.6 [17] protocol for assembly. To enhance the accuracy of whole-genome sequencing, the Illumina iSeq 100 (Illumina, San Diego, CA, USA) contributed to the resequencing of IDCC 3401 genomic data. The BWA-MEM algorithm [18] facilitated the mapping of 150-base-pair paired-end reads from the iSeq 100 system, and the GATK haplotypecaller corrected single-base errors [19]. The final genome assembly was annotated through rapid annotation using subsystem technology [20]. The average nucleotide identity (ANI) value with the most similar strains was calculated using an ANI calculator (Kostas Lab, Atlanta, GA, USA). After annotating the open reading frames of the genome, the functional genes were analyzed using eggNOG-mapper v2.

To investigate the genetic safety of *L. paracasei* IDCC 3401, virulence genes were searched using the BLASTn algorithm with the VFDB (http://www.mgc.ac.cn/VFs/ (accessed on 1 November 2023); thresholds for identity > 80%, coverage > 70%, and E-value < 1 × 10^−5^) [21]. In addition, antibiotic resistance genes were analyzed by comparing the assembled sequences with the reference sequences in the ResFinder database (https://cge.cbs.dtu.dk/services/ResFinder/) (accessed on 1 November 2023) using the ResFinder 4.1 software [22]. The search parameters for the analysis were sequence identity > 80% and coverage > 60%.

### 2.3. Determination of Minimum Inhibitory Concentrations

The susceptibility of *L. paracasei* IDCC 3401 to a variety of antibiotics typically used to treat enterococcal infections was investigated using the MIC test. Nine antibiotics were selected: ampicillin, vancomycin, gentamicin, kanamycin, streptomycin, erythromycin, clindamycin, tetracycline, and chloramphenicol (Sigma-Aldrich, St. Louis, MO, USA). The test was conducted based on the Clinical and Laboratory Standards Institute (CLSI) guidelines [23]. Briefly, a single colony of *L. paracasei* IDCC 3401 was transferred into MRS broth and preincubated for 16–18 h. The cultured cells and antibiotic solution were mixed in a 96-well plate to obtain the initial cell density of 5 × 10^5^ CFU/mL and antibiotic concentration of 0.125–1024 µg/mL. After incubation at 37 °C for 18–20 h, optical density was measured using a microplate reader (EPOCH2, BioTek). Determination of resistance and susceptibility against each antibiotic followed the EFSA guidelines [24].

### 2.4. Biogenic Amine and L-/D-Lactate Production

To examine the BA production of *L. paracasei* IDCC 3401, a previously described method [25] was used with minor modifications. Briefly, the supernatants were obtained via centrifugation at 6000 rpm and at 4 °C for 30 min and filtered through a 0.20 μm membrane filter. To extract the BAs, 0.5 mL of 0.1 M HCl was mixed with 0.5 mL of supernatants, followed by filtering through a 0.45 μm membrane filter. After incubating 1 mL of the extracted BAs in a water bath at 70 °C for 10 min, derivatization was performed by adding 200 μL of saturated NaHCO_3_, 20 μL of 2 M NaOH, and 0.5 mL of dansyl chloride (10 mg/mL acetone). The derivatized BAs were mixed with 200 μL of L-proline (100 mg/mL H_2_O) and incubated in the dark at room temperature for 15 min. Acetonitrile (high-performance liquid chromatography (HPLC)-grade; Sigma-Aldrich, St. Louis, MO, USA) was added to bring the final volume of the mixture to 5 mL. Finally, the BAs were separated and quantified via HPLC; Agilent 1260, Agilent Technologies, Santa Clara, CA, USA) with a C18 column (YMC-Triart, 4.6 × 250 mm, YMC, Kyoto, Japan). Aqueous acetonitrile solution (acetonitrile: H_2_O = 67:33 *v*/*v*) was used as a mobile phase at a constant flow rate of 0.8 mL/min. Peaks were detected at 254 nm using a UV detector (G7115A, Agilent Technologies) and quantified according to each calibration curve of the BAs (Sigma-Aldrich) such as tyramine, histamine, putrescine, 2-phenethylamine, cadaverine, and tryptamine.

To quantify L-/D-lactate in the supernatants of *L. paracasei* IDCC 3401, an assay was performed using a Megazyme kit according to the manufacturer’s instructions (Bray, Ireland). Briefly, 0.1 mL of the supernatants of bacterial culture was mixed with 1.5 mL of H_2_O, 0.5 mL of supplied buffer (pH 10.0), 0.1 mL of NAD^+^ solution, and 0.02 mL of glutamate–pyruvate transaminase. The reaction mixture was incubated at room temperature for 3 min. Then, D-lactate absorbance was measured at 340 nm. Next, 0.02 mL of 2000-U/mL lactate dehydrogenase was added to the above reaction mixture, and L-/D-lactate absorbance was measured at 340 nm until the LD reaction stopped. Finally, L-/D-lactate concentrations were calculated according to the equation provided by the manufacturer.

### 2.5. Single-Dose Acute Oral Toxicity

According to the OECD guidelines for testing chemicals [26], a single-dose acute oral toxicity test (No. TGK-2022-000580) was conducted at the Korea Testing and Research Institute (Hwasun, Republic of Korea). Briefly, 12 female rats were divided into 4 groups: 2 groups of 9-week-old rats and 2 groups of 10-week-old rats. The rats were kept under the following conditions: 12 h light/dark cycle, 150–300 lux of illumination, temperature of 19.9 °C–22.6 °C, and relative humidity of 37.4–60.2%. Each group was orally administered 300 or 2000 mg of lyophilized *L. paracasei* IDCC 3401 in 10 mL of sterilized water. Over 14 days, clinical signs, body weight changes, and necropsy findings were examined. A Student’s *t*-test was used to analyze the differences between time points, and *p* < 0.05 was considered significant.

### 2.6. Acid and Bile Tolerance

The tolerance of *L. paracasei* IDCC 3401 in low pH and bile salt was examined using a previously described method [27] with a slight modification. The strain was cultured overnight in MRS broth. For the acid tolerance assay, the cultures (1%) were inoculated into sterilized MRS broth with an adjusted pH of 2.5 using 1 M HCL solution. After incubation for 2 or 4 h at 37 °C, the *L. paracasei* IDCC 3401 culture was serially diluted in and plated on MRS agar. To examine bile tolerance, overnight cultures (1%) were incubated in MRS broth containing 0.3% (*w*/*v*) bile salt (OX gall/OX bile) for 2 or 4 h. Then, the viability of the *L. paracasei IDCC* 3401 was determined by plating on MRS agar. The viability of *L. paracasei IDCC* 3401 was expressed as a percentage of survival cells after incubation compared with the initial cell number.

### 2.7. Carbohydrate Utilization Patterns and Enzymatic Activities

The carbohydrate utilization patterns of *L. paracasei* IDCC 3401 were investigated using the API 50 CHL/CHB Kit (BIOMÉRIUX, Marcy-l’Étoile, France) containing 49 different types of carbohydrates according to the manufacturer’s protocol. *L. paracasei* IDCC 3401 was anaerobically cultured in MRS broth at 37 °C for 24 h. After centrifugation, the bacterial pellet was resuspended in 10 mL of API 50 CHL medium. Subsequently, the suspension (6.0 × 10^8^ CFU/mL) was inoculated into cupules containing each carbohydrate and incubated at 37 °C for 48 h. Carbohydrate fermentation patterns were evaluated by observing color changes according to the manufacturer’s instructions.

Enzymatic activities of *L. paracasei* IDCC 3401 were determined using the API ZYM Kit (bioMerieux Inc., Marcy l’Etoile, France) against 19 different enzymes according to the manufacturer’s protocol. The bacterial pellet, which was collected as previously described, was suspended in phosphate-buffered saline (PBS) (pH 7.4). The suspension (1.8 × 10^9^ CFU/mL) was inoculated into cupules containing each substrate and incubated at 37 °C for 4 h. Then, one drop of the ZYM-A and ZYM-B reagents was added to each cupule. After reacting for 5 min at room temperature, color changes in the mixture were observed to determine enzymatic activities.

### 2.8. Antipathogenic Activities

To examine the antipathogenic effects of *L. paracasei* IDCC 3401, a previously described method [28] was used with a minor modification. The supernatants of *L. paracasei* IDCC 3401 were harvested via centrifugation at 8000 rpm and 4 °C for 30 min, followed by filtering through a 0.20 µm syringe filter to remove residual bacteria. Four pathogenic strains, namely, *Staphylococcus aureus* ATCC 25923, *Enterococcus faecalis* ATCC 29212, *Bacillus cereus* ATCC 14579, and *Salmonella* Typhimurium ATCC 13311, were adjusted to the initial cell density of 1.5 × 10^8^ CFU/mL using the McFarland standard (bioMerieux, Marcy-I’Etoile, France). Then, 100 μL of pathogenic bacteria culture was incubated at 37 °C with 100 μL of supernatants of *L. paracasei* IDCC 3401 in a 96-well culture plate. Finally, an optical density of 595 nm was measured using a microplate reader (BioTek, Winooski, VT, USA) at 0 and 24 h.

### 2.9. Adhesion Assay

To explore the ability of *L. paracasei* IDCC 3401 to adhere to host cells, a previously described method [29] was used with few modifications. The human intestinal epithelial cell line HT-29 cells were cultured in Dulbecco’s modified Eagle’s medium (DMEM; Welgene, Gyeongsan, Korea) supplemented with 10% fetal bovine serum and 10% penicillin/streptomycin (HyClone, Logan, UT, USA) at 37 °C in a 5% CO_2_ humidified incubator. HT-29 cells (5 × 10^5^ cells/mL) were seeded in a culture dish and incubated until the cells formed a monolayer. The cells were washed with PBS and treated with bacterial suspension (1 × 10^8^ CFU/mL) in antibiotic-free DMEM at 37 °C for 2 h. Subsequently, the cells were lysed in 0.05% trypsin/EDTA at 37 °C for 20 min. To determine bacterial adherence, cell lysates were serially diluted and enumerated by plating on MRS agar. Adhesion was expressed as the percentage of bacteria adherent to HT-29 cells.

## 3. Results

### 3.1. Genetic Analysis of L. paracasei IDCC 3401

As a result of genetic analysis, the target strain in this study was identified as *Lacticaseibacillus paracasei* (formerly known as *Lactobacillus paracasei*) showing the highest similarity based on ANI values. As presented in Appendix A, the genome has 2,995,875 base pairs with 46.59% GC content and 2983 predicted coding DNA sequences. The functional annotation indicated that carbohydrate transport and metabolism (G, 12.99%) and amino acid transport and metabolism (E, 11.73%) were among the essential functions in *L. paracasei* (Appendix A). These results suggest that *L. paracasei* IDCC 3401 degrades a wide range of proteins and carbohydrates for adaptation in various environments and facilitates the absorption of nutrients in the host.

To explore the potential toxigenic effects of *L. paracasei* IDCC 3401, genes associated with virulence and antibiotic resistance were identified via in silico analysis. As presented in Appendix A, no resistance genes were identified in the genome and plasmid sequences. Furthermore, no virulence genes were found in the strain. These results suggest that *L. paracasei* IDCC 3401 is safe based on our genomic evaluation.

### 3.2. Antibiotic Susceptibility

The expression of antibiotic resistance in *L. paracasei* IDCC 3401 is an important safety criterion as the antibiotic resistance of probiotics can be transferred to commensal bacteria or pathogens [30]. *L. paracasei* IDCC 3401 was found to be susceptible to all the antibiotics tested, except for kanamycin (Table 1). However, kanamycin resistance is generally observed in most lactic acid bacteria including lactobacillus, and thus it can be deduced that the *L. paracasei* IDCC 3401 strain does not exhibit antibiotic resistance.

### 3.3. Toxic Compound Production

Because BA or D-lactate produced by bacteria poses health risks to humans [30], the ability of *L. paracasei* IDCC 3401 to produce these toxic compounds was analyzed in the supernatants (Table 2). BAs such as tyramine, histamine, putrescine, 2-phenethylamine, cadaverine, and tryptamine were investigated but were not detected in the supernatants. The ratios of L-lactate and D-lactate were determined to be 94.85% and 5.15%, respectively. These results suggest that *L. paracasei* IDCC 3401 is safe for BA and D-lactate production.

### 3.4. Single-Dose Acute Oral Toxicity Study in Rats

Investigating the toxicity of probiotics is important, and oral administration to animals is a prerequisite for the use of newly isolated bacteria as a probiotic strain. As presented in Table 3, no body weight reduction was observed in any of the administration groups. Furthermore, there was no mortality or clinical signs during the study period. For necropsy, *L. paracasei* IDCC 3401 did not exert abnormal effects in rats. These results suggest that *L. paracasei* IDCC 3401 does not exhibit toxicity for consumption under the conditions in this study.

### 3.5. Viability under Acid and Bile Stresses

The acid tolerance and bile tolerance of *L. paracasei* IDCC 3401 were measured as general probiotic properties as the viability of probiotics needs to be maintained during and after consumption [32]. In a pH 2.5 medium, the viability of the strain was slightly reduced to 96% and 94% after incubation for 2 and 4 h, respectively (Figure 1A). Similarly, the viability of the strain was marginally reduced by 98% and 96% in the presence of 0.3% bile salt after incubation for 2 and 4 h, respectively (Figure 1B). These results suggest that *L. paracasei* IDCC 3401 exhibits tolerance against acid and bile stresses.

### 3.6. Carbohydrate Utilization and Enzymatic Activities

Carbohydrate utilization and enzymatic activities were investigated because probiotics can utilize various carbohydrates for their proliferation and metabolism in the intestine [33]. Among 49 carbohydrates, *L. paracasei* IDCC 3401 utilized 17 carbohydrates: ribose, galactose, d-glucose, d-fructose, d-mannose, mannitol, n-acethyl-glucosamine, esculine, salicine, cellobiose, lactose, trehalose, melizitose, gentiobiose, d-turanose, d-tagatose, and gluconate (Table 4).

The enzymatic activity of *L. paracasei* IDCC 3401 against 19 enzymes responsible for carbohydrate, lipid, and vitamin metabolism was investigated. As shown in Table 5, the strain had various enzymatic activities, including esterase, esterase lipase, leucine arylamidase, valine arylamidase, cystine arylamidase, acid phosphatase, alkaline phosphatase, naphthol-AS-BI-phosphohydrolase, α-galactosidase, and β-glucosidase. β-glucuronidase activity, however, was negative in *L. paracasei* IDCC 3401.

### 3.7. Antipathogenic Activities against Enteric Pathogens

To positively change the gut microbiome via the colonization of probiotics, the growth of pathogens should be inhibited by probiotics [34]. To explore antimicrobial activities, enteric pathogenic microorganisms were cultured with the supernatants of *L. paracasei* IDCC 3401. *Lacticaseibacillus rhamnosus* GG (LGG), which is widely used as a commercial probiotic strain, was used as a positive control. As shown in Figure 2, the growth of pathogens cultured with supernatants of *L. paracasei* IDCC 3401 exhibited significant differences compared with the culture of pathogens alone. The growth rates of *S. aureus* ATCC 25923, *E. faecalis* ATCC 29212, *B. cereus* ATCC 14579, and *S. typhimurium* ATCC 13311 decreased by 21%, 23%, 14%, and 27%, respectively, compared with the control without supernatants of *L. paracasei* IDCC 3401. Growth inhibition by *L. paracasei* IDCC 3401 was similar or superior to that by LGG. These results suggest that *L. paracasei* IDCC 3401 inhibits the growth of pathogens in the gastrointestinal tract.

### 3.8. Adhesion to HT-29 Cell Line

For probiotics to competitively adhere to the mucosa, it is important to examine the adhesion and colonization of probiotics in intestinal epithelial cells [35]. Figure 3 demonstrates that *L. paracasei* IDCC 3401 exhibited approximately 91.8% adhesion ability to HT-29 cells but did not significantly differ from LGG, which exhibited approximately 93.4% adhesion ability. *L. paracasei* IDCC 3401 showed a strong ability to adhere to intestinal cells in our in vitro assay.

## 4. Discussion

In this study, genomic and phenotypic analyses of *L. paracasei* IDCC 3401 were conducted to explore the safety and general probiotic properties of the strain. Newly isolated strains to be used as probiotics should be identified at the whole-genome level to confirm the characteristics of the raw material; their safety should also be examined by evaluating antibiotic resistance, toxin production, and metabolic characteristics according to the guidelines for the evaluation of probiotics in food, such as the FAO/WHO guidelines (2002) [36]. The MIC test revealed that the strain identified as *L. paracasei* IDCC 3401 was susceptible to all antibiotics, except for kanamycin, and that it did not exhibit antibiotic resistance and virulence genes. Therefore, *L. paracasei* IDCC 3401 does not have the ability to transfer antibiotic-resistant genes to pathogens [37]. Furthermore, most *Lacticaseibacillus* species are intrinsically resistant to aminoglycosides, such as kanamycin, and this intrinsic resistance is not regarded as transferable to commensal microbes [38].

Some LAB species can produce BAs through the decarboxylation of amino acids during food fermentation [39]. However, BAs produced by LAB in foods are undesirable for safety. Specifically, consumption of BAs at high concentrations may cause headaches, respiratory distress, heart palpitation, hyper- or hypotension, and several allergenic disorders [40]. In the present study, because supernatants of *L. paracasei* IDCC 3401 were not found to contain BAs, the strain can be safely used as a food material [41]. Furthermore, lactate, produced from pyruvate through the metabolism of carbohydrates by LAB, has two optical isomers, L- and D-types [42]. In the human body, L-lactate is only metabolized because of the absence of D-lactated dehydrogenase for D-lactate production [43]. D-lactate is mainly a consequence of bacterial production, which accumulates in the blood, thus resulting in D-lactic acidosis [44]. Because D-lactate production by *L. paracasei* IDCC 3401 was significantly lower than L-lactate production, *L. paracasei* IDCC 3401 exhibited positive metabolic properties associated with the production of toxic substances. In addition, *L. plantarum* CRD7 and *L. rhamnosus* CRD11 produced more D- lactic acid (0.65 ± 0.13 g/L and 0.70 ± 0.38 g/L, respectively) than *L. paracasei* IDCC 3401 (0.41 ± 0.05 g/L) [45]. Single-dose acute oral toxicity tests in rats can provide physiologic information as evidence of safety for human consumption [46]. Clinical signs, body weight changes, and necropsy findings were not exhibited by rats administered *L. paracasei* IDCC 3401.

The viability of probiotics has been recognized as an essential factor to ensure beneficial health effects on the host. It assumes that the probiotic strain remains viable during passage through the gastrointestinal tract despite facing various stressors such as low pH and the presence of bile acid. In this study, the viability of *L. paracasei* IDCC 3401 was maintained above 90% at low pH and with bile acid. Compared with previous studies [47,48], *L. paracasei* IDCC 3401 exhibited higher survival rates than other LAB under conditions similar to those in our study. These results indicate that *L. paracasei* IDCC 3401 exhibits strong tolerance against acid and bile as it survived and proliferated in the human body and further maintained probiotic activity. The utilization of various carbohydrates is associated with the adaptation ability of bacteria in different environments [49]. Through the fermentation of carbohydrates, LAB generate health-beneficial secondary metabolites, particularly short-chain fatty acids [50]. *L. paracasei* IDCC 3401 can survive in the intestine by utilizing various carbohydrates associated with galactose metabolism (D-glucose, D-galactose, D-mannose, and D-fructose) and lactose degradation (D-glucose, D-galactose) (Appendix A), which were based on SMPDB and KEGG. Furthermore, bacterial β-glucuronidase is a potentially important enzyme in the generation of toxic compounds associated with colorectal cancer [51]. However, β-glucuronidase was not detected in *L. paracasei* IDCC 3401, thus proving its safety. *L. paracasei* CNCM I-4034 in another study was comparable to *L. paracasei* IDCC 3401 in β-glucuronidase activity [52]. Enteric pathogens, including *S. aureus*, *E. faecalis*, *B. cereus*, and *S. typhimurium*, can induce gastrointestinal problems [53]. Probiotic strains are known to inhibit the growth of intestinal pathogens by secreting antimicrobial substances such as bacteriocins, lactic acid, hydrogen peroxide, and other composites [54]. The present study demonstrated that *L. paracasei* IDCC 3401 significantly inhibited the growth of the four enteric pathogens. This result was comparable to that of the CFS of *L. rhamnosus* NCDC953 and *L. reuteri* NCDC958 inhibiting the growth of *Bacillus cereus*, *Salmonella typhimurium*, *Enterococcus faecalis*, and *Staphylococcus aureus* [55]. The adhesion of bacteria to the human intestinal mucosa, which is influenced by cell surface components, is one of the crucial properties of probiotics to achieve successful colonization in gastrointestinal environments [56]. In another study, the probiotic strain *L. paracasei* LBC-81 exhibited lower adhesion ability than *L. paracasei* IDCC 3401 [56]. Furthermore, the adhesion ability of LAB was reported to range from 20% to 90% [57]. Comparable to these results, *L. paracasei* IDCC 3401 exhibited a strong ability to adhere to intestinal cells.

*Lactobacillus paracasei* is used as a microbial food culture (MFC) and probiotic [58]. It is also known to be used as a starter culture for milk fermentation, promoting flavor development in dairy products and improving functional properties in meat products [22]. In this study, the safety and characteristics of *L. paracasei* IDCC 3401 as a probiotic were confirmed, and it is expected that it will be possible to use it as a probiotic health functional food in the future.

## 5. Conclusions

This study demonstrated the safety and characteristics of *L. paracasei* IDCC 3401. Both genomic and phenotypic analyses revealed the safety of *L. paracasei* IDCC 3401, which had no antibiotic resistance issues, no production of toxic substances in vitro and in vivo, and potential survivability in the intestine. Although further clinical studies are warranted to confirm the health risks posed by *L. paracasei* IDCC 3401, these results suggest that it can be considered safe for use as a probiotic.

## Figures and Tables

**Figure 1 microorganisms-12-00085-f001:**
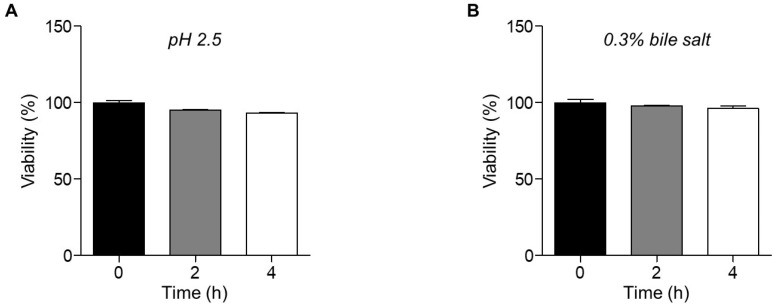
Viability of *L. paracasei* IDCC 3401 in the presence of (**A**) acidic stress (pH = 2.5) and (**B**) 0.3% bile salt for 2 and 4 h. Experimental data are expressed as mean ± standard deviations of three independent experiments. Statistical significance between the groups was determined via analysis of variance (ANOVA) when *p* < 0.05.

**Figure 2 microorganisms-12-00085-f002:**
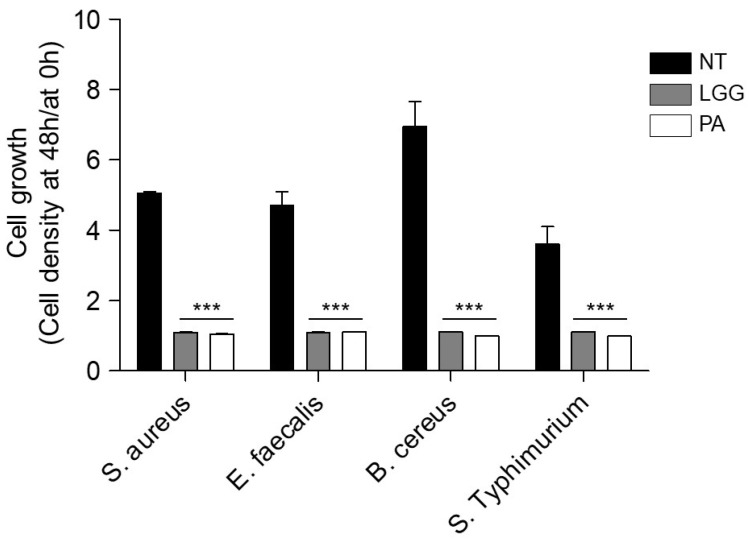
Antipathogenic activities of *L. paracasei* IDCC 3401 against enteric pathogens, including *S. aureus* ATCC 25923, *E. faecalis* ATCC 29212, *B. cereus* ATCC 14579, and *S. typhimurium* ATCC 13311. Experimental data are expressed as means ± standard deviations of three independent experiments. *** on the bar showed statistical significance between the groups determined via ANOVA when *p* < 0.05. N.T., not treated; LGG, *L. rhamnosus* GG; PA, *L. paracasei* IDCC 3401.

**Figure 3 microorganisms-12-00085-f003:**
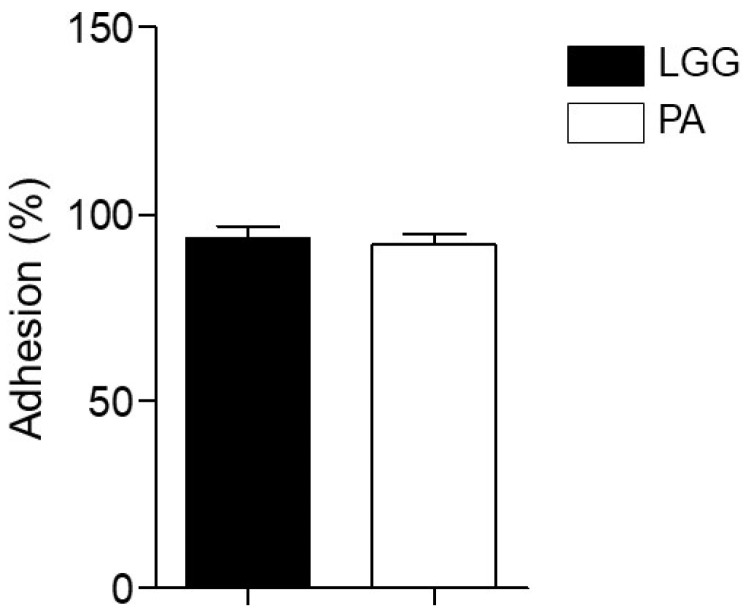
Adhesion of *L. paracasei* IDCC 3401 to HT-29 cells. Experimental data are expressed as means ± standard deviations of three independent experiments. Statistical significance between the groups was determined via ANOVA when *p* < 0.05. LGG, *L. rhamnosus* GG; PA, *L. paracasei* IDCC 3401.

**Table 1 microorganisms-12-00085-t001:** Minimal inhibitory concentrations of *L. paracasei* IDCC 3401.

Items	Antibiotics ^5^								
	AMP	VAN	GEN	KAN	STR	ERY	CLI	TET	CHL
Cutoff value (µg/mL) ^1^	4	n.r. ^4^	32	64	64	1	4	4	4
Observed MIC	0.5–1	512>	32	256	64	<0.125	<0.125	1	2
Assessment	S ^2^	n.r.	S	R ^3^	S	S	S	S	S

^1^ EFSA [31]. ^2^ S, susceptible. ^3^ R, resistant. ^4^ n.r., not required. ^5^ Abbreviations: AMP, ampicillin; VAN, vancomycin; GEN, gentamicin; KAN, kanamycin; STR, streptomycin; ERY, erythromycin; CLI, clindamycin; TET, tetracycline; CHL, chloramphenicol.

**Table 2 microorganisms-12-00085-t002:** Production of biogenic amine and L-/D-lactate by *L. paracasei* IDCC 3401.

Biogenic Amine (mM)					
Tyramine	Histamine	Putrescine	2-Phenethylamine	Cadaverine	Tryptamine
n.d. ^a^	n.d.	n.d.	n.d.	n.d.	n.d.
D-/L-lactate proportion					
L-lactate (g/L)		D-lactate (g/L)		L-form (%)	D-form (%)
7.59 ± 0.98		0.41 ± 0.05		94.85	5.15

^a^ n.d.: not detected.

**Table 3 microorganisms-12-00085-t003:** Body weight changes in rats administered *L. paracasei* IDCC 3401.

Group	Dose (g/kg BW ^1^)	Day after Administration				
		0	1	3	7	14
9 weeks old	300	202.1 ± 11.9	222.0 ± 12.2	235.8 ± 9.3	246.9 ± 9.9	258.0 ± 8.6
	2000	219.5 ± 15.4	240.7 ± 17.2	249.7 ± 18.6	262.5 ± 19.8	273.8 ± 24.8
10 weeks old	300	232.5 ± 13.2	257.1 ± 15.4	263.7 ± 12.5	275.5 ± 15.1	286.4 ± 5.1
	2000	231.0 ± 19.6	254.5 ± 23.6	266.5 ± 21.3	273.6 ± 24.3	281.7 ± 28.9

^1^ BW, body weight.

**Table 4 microorganisms-12-00085-t004:** Carbohydrate fermentation patterns of *L. paracasei* IDCC 3401.

Substrate	Result ^a^	Substrate	Result	Substrate	Result
Glycerol	–	Mannitol	+	D-Raffinose	–
Erythritol	–	Sorbitol	–	Amidon	–
D-Arabinose	–	α-Methyl-D-mannoside	–	Glycogen	–
L-Arabinose	–	α-Methyl-D-glucoside	–	Xylitol	–
Ribose	+	N-Acetyl-Glucosamine	+	Gentibiose	+
D-Xylose	–	Amygdaline	–	D-Turanose	+
L-Xylose	–	Arbutine	–	D-Lyxose	–
Adonitol	–	Esculine	+	D-Tagatose	+
β-Methyl-xylose	–	Salicine	+	D-Fucose	–
Galactose	+	Cellobiose	+	L-Fucose	–
D-Glucose	+	Maltose	–	D-Arabitol	–
D-Fructose	+	Lactose	+	L-Arabitol	–
D-Mannose	+	Melibiose	–	Gluconate	+
L-Sorbose	–	Sucrose	–	2-Keto-gluconate	–
Rhamnose	–	Trehalose	+	5-Keto-gluconate	–
Dulcitol	–	Inuline	–		
Inositol	–	Melizitose	+		

^a^ +: carbohydrate utilization; –: no carbohydrate utilization.

**Table 5 microorganisms-12-00085-t005:** Enzyme activities of *L. paracasei* IDCC 3401.

Enzyme	Activity	Enzyme	Activity
Alkaline phosphatase	+ ^a^	Naphthol-AS-BI-phosphohydrolase	+
Esterase	+	α-Galactosidase	+
Esterase Lipase	+	β-Galactosidase	–
Acid phosphatase	+	α-Glucosidase	–
Lipase	– ^a^	β-Glucosidase	+
Leucine arylamidase	+	β-Glucuronidase	–
Valine arylamidase	+	N-Acetyl-β-glucosaminidase	–
Cystine arylamidase	+	α-Mannosidase	–
Trypsin	–	α-Fucosidase	–
α-Chymotrypsin	–		

^a^ +: enzyme activity; –: no enzyme activity.

## Data Availability

Data are contained within the article and Appendix A.

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
