# Peer review of "Isolation, Characterization, and Safety Evaluation of the Novel Probiotic Strain Lacticaseibacillus paracasei IDCC 3401 via Genomic and Phenotypic Approaches"

_microorganisms, 2023, doi:10.3390/microorganisms12010085_

Round 1

Reviewer 1 Report

Comments and Suggestions for Authors

The work of Lee et al. is devoted to the characterization of a new probiotic strain Lacticaseibacillus paracasei IDCC 3401 by performing a set of tests of a fairly wide range in order to prove its harmlessness. It is a pity that the paper lacks the main and key test that would prove that the strain does eliminate the pathological microbiome by replacing it (or discussions of why it shouldn't/wasn't done). There are no experiments to analyze rat feces for the presence of this strain during feeding. In my opinion, the absence of these experiments is a direct obstacle to the publication of the work.

My line-by-line comments:

Line 69-96: no information on the source of strain IDCC 3401. By whom isolated? From where? Where is it stored? Why selected? -- these characteristics are necessary because the words "isolation" and "characterization" appear in the title.

Line 82: why do the authors use a whale designed to isolate DNA from blood!

Line 84: no information about obtaining libraries, sequencing format: paired-end reids, lengths, etc. How was the genome assembled? How was the assembly performed? How was the data analyzed? Where was the genomic data deposited? Were reads deposited? How were the reads annotated?

Lines 90-393: here and below there are no references to the software used in the paper.

Lines 98-393: here and below missing references to methods used in the paper (not always, but mostly).

Line 144: Is the use of 12 individual rats sufficient for the study? No description of the methods of statistical processing. What criteria were used to prove the significance of the results? A paragraph on data analysis should be added to the text.

Line 151: there is no reference in this paragraph. Where did the methodology come from?

Line 162: same as in the previous comment

Line 180: same as in the previous comment

Table 5: typo in "L-form" instead of "D-form"

Table 6: no control information available. What happened to the rats that did not take IDCC 3401? I would like to see information on both groups of 9-week-old rats and 10-week-old rats with a dose of "0".

Lines 206-332: The results section also needs to demonstrate that the strain is effective as a probiotic? Does it displace other strains in the microbiota? Is it detectable in the feces when fed or otherwise inoculated? It would also be good to see the growth parameters of the strain, its growth rates, and doubling periods. If it is replacing other strains it should appear in the microbiota, over what time period? In what ratio? Are its live bacteria in the rat feces?

Lines 334-393: no comparative information with other probiotics, discuss growth rates, pathogen replacement, etc. On what basis do the authors believe the strain will behave like a probiotic under REAL conditions?

Author Response

Ms. Ref. No.: microorganisms-2771755

Title: Isolation, characterization, and safety evaluation of the novel probiotic strain Lacticaseibacillus paracasei IDCC 3401 via genomic and phenotypic approaches

Reviewers' comments:

Reviewer #1: The work of Lee et al. is devoted to the characterization of a new probiotic strain Lacticaseibacillus paracasei IDCC 3401 by performing a set of tests of a fairly wide range in order to prove its harmlessness. It is a pity that the paper lacks the main and key test that would prove that the strain does eliminate the pathological microbiome by replacing it (or discussions of why it shouldn't/wasn't done). There are no experiments to analyze rat feces for the presence of this strain during feeding. In my opinion, the absence of these experiments is a direct obstacle to the publication of the work.

Response: We thank the reviewer for the overall comments. We have divided and answered for each comment made by reviewer as below.

  1. Line 69-96: no information on the source of strain IDCC 3401. By whom isolated? From where? Where is it stored? Why selected? -- these characteristics are necessary because the words "isolation" and "characterization" appear in the title.

Response: We thank the reviewer’ comments. As suggested by the reviewer, we have added the information. Lacticaseibacillus paracasei IDCC 3401 (ATCC No. BAA-2839) was isolated from breast milk-fed infant feces in 2005 by Ildong Pharmaceutical Co., Ltd, and stored in Ildong Bioscience. The information on this strain has already been registered in ATCC. Breast-feeding babies have a healthy microbiome, so L. paracasei IDCC 3401 newly isolated from them was selected as a probiotic candidate.

L69, Pages 2: L. paracasei IDCC 3401 (ATCC No. BAA-2839) was isolated from the feces of breast milk-fed infants by Ildong Bioscience.

  1. Line 82: why do the authors use a whale designed to isolate DNA from blood!

Response: As reviewer found error in the manuscript, we corrected it as follows.

L80-81, Pages 2: Genomic DNA of L. paracasei IDCC 3401 was extracted using a Wizard Genomic DNA Purification Kit (Promega Co., Madison, WI, USA) according to the manufacturer’s instructions.

  1. Line 84: no information about obtaining libraries, sequencing format: paired-end reids, lengths, etc. How was the genome assembled? How was the assembly performed? How was the data analyzed? Where was the genomic data deposited? Were reads deposited? How were the reads annotated?

Response: Thanks for review’s comments. As kindly suggested, we corrected the manuscript as follows.

L82-89, Pages 2: The genome of strain IDCC 3401 underwent comprehensive sequencing utilizing both Pacific Biosciences (PacBio) RS II and Illumina iSeq 100 systems. PacBio RS II data were processed using the Canu v1.6 [17] protocol for assembly. To enhance the accuracy of whole genome sequencing, Illumina iSeq 100 contributed to the resequencing of IDCC 3401 genomic data. The BWA-MEM algorithm [18] facilitated the mapping of 150-base pair paired-end reads from the iSeq 100 system, and GATK haplotypecaller corrected single base errors [19]. The final genome assembly was annotated through the Rapid Annotation using Subsystem Technology [20].

  1. Lines 90-393: here and below there are no references to the software used in the paper.

Response: We thank the reviewer for the suggestion. As suggested by the reviewer, we have added the references.

L93-99, Pages 3 : To investigate the genetic safety of L. paracasei IDCC 3401, virulence genes were searched using the BLASTn algorithm with the VFDB (http://www.mgc.ac.cn/VFs/; thresholds for identity > 80%, coverage > 70%, and E-value < 1E-5) [21]. In addition, antibiotic resistance genes were analyzed by comparing the assembled sequences with the reference sequences in the ResFinder database (https://cge.cbs.dtu.dk/services/ResFinder/) using the ResFinder 4.1 software [22].

  1. Lines 98-393: here and below missing references to methods used in the paper (not always, but mostly).

Response: We thank the reviewer for the suggestion. As suggested by the reviewer, we have added the references.

L105-107, Pages 3 : The test was conducted based on the Clinical and Laboratory Standards Institute(CLSI) guidelines [23].

L115-116, Pages 3 : To examine the BA production of L. paracasei IDCC 3401, a previously described method [25] was used with minor modifications.

  1. Line 144: Is the use of 12 individual rats sufficient for the study? No description of the methods of statistical processing. What criteria were used to prove the significance of the results? A paragraph on data analysis should be added to the text.

Response: We thank the reviewer for the suggestion. As suggested by the reviewer, we have added the statistical analysis. To perform triplicate experiments, 4 groups with 3 rats in each group as age and administrated dose were tested over a period. There are no significant differences at day 1, 1, 3, 7, and 14 after the administration compared to day 0 (before administration)

L152-153, Pages 4 : A student’s t-test was used to analyze the differences between time points, and p < 0.05 was considered significant.

  1. Line 151: there is no reference in this paragraph. Where did the methodology come from?

Response: We thank the reviewer for the suggestion. As suggested by the reviewer, we have added the reference.

L156-157, Pages 4: The tolerance of L. paracasei IDCC 3401 in low pH and bile salt was examined as a previously described method [27] with a slight modification.

  1. Line 162: same as in the previous comment

Response: We thank the reviewer for the suggestion. As suggested by the reviewer, we added the methods.

L170, Pages 4 : The carbohydrate utilization patterns of L. paracasei IDCC 3401 were investigated using the API 50 CHL/CHB Kit (BIOMÉRIUX, Marcy-l’Étoile, France) containing 49 different types of carbohydrates according to the manufacturer’s protocol.

L177-178, Pages 4 : Enzymatic activities of L. paracasei IDCC 3401 were determined using the API ZYM Kit (bioMerieux Inc., Marcy l’Etoile, France) against 19 different enzymes according to the manufacturer’s protocol.

  1. Line 180: same as in the previous comment

Response: We thank the reviewer for the suggestion. As suggested by the reviewer, we have added the references.

L186-187, Pages 4 : To examine the antipathogenic effects of L. paracasei IDCC 3401, a previously described method [28] was used with a minor modification.

L199-200, Pages 4 : To explore the ability of L. paracasei IDCC 3401 to adhere to host cells, a previously described method [29] was used with few modifications.

  1. Table 5: typo in "L-form" instead of "D-form"

Response: We thank the reviewer for the suggestion. As suggested by the reviewer, we corrected it.

Table 2, Pages 6

Table 2. Production of biogenic amine and L-/D-lactate by L. paracasei IDCC 3401

Biogenic amine (mM)

Tyramine

Histamine

Putrescine

2-Phenethylamine

Cadaverine

Tryptamine

n.d.a

n.d.

n.d.

n.d.

n.d.

n.d.

D-/L-lactate proportion

L-lactate (g/l)

D-lactate (g/L)

L-form (%)

D-form (%)

7.59 ± 0.98

0.41 ± 0.05

94.85

5.15

a n.d.: not detected.

  1. Table 6: no control information available. What happened to the rats that did not take IDCC 3401? I would like to see information on both groups of 9-week-old rats and 10-week-old rats with a dose of "0".

Response: We thank the reviewer for the suggestion. The value at Day 0 as control was measured just before the administration of L. paracasei IDCC 3401 in rats.

  1. Lines 206-332: The results section also needs to demonstrate that the strain is effective as a probiotic? Does it displace other strains in the microbiota? Is it detectable in the feces when fed or otherwise inoculated? It would also be good to see the growth parameters of the strain, its growth rates, and doubling periods. If it is replacing other strains it should appear in the microbiota, over what time period? In what ratio? Are its live bacteria in the rat feces?

Response: We thank the reviewer for the suggestion. Our study determined whether a novel strain possesses the safety and characteristics as a probiotic candidate in in vitro. As a result, we confirmed that L. paracasei IDCC 3401 can be used as a safe food material and possesses tolerance in intestinal conditions, antimicrobial activity, and superior adhesion ability on intestinal cell lines. Based on this study, we plan to perform further studies to confirm the probiotic effects of L. paracasei IDCC 3401 such as microbiota improvement in in vivo and clinical studies.

  1. Lines 334-393: no comparative information with other probiotics, discuss growth rates, pathogen replacement, etc. On what basis do the authors believe the strain will behave like a probiotic under REAL conditions?

Response: We thank the reviewer for the suggestion. As suggested by the reviewer, we have added the comparative information with other probiotics. In this study, we predicted the behavior of this strain by performing experiments under conditions similar to the intestinal environment using acid, bile salts, intestinal pathogens, and intestinal cell lines in in vitro. As answered previously, we plan to perform further studies to confirm the probiotic effects of L. paracasei IDCC 3401 such as microbiota improvement under real conditions in in vivo and clinical studies.

L357-360, Pages 10 : In addition, L. plantarum CRD7 and L. rhamnosus CRD11 more produced D- lactic acid (0.65 ± 0.13 g/L and 0.70 ± 0.38 g/L, respectively) than L. paracasei IDCC 3401 (0.41 ± 0.05 g/L) [44].

L380-382, Pages 10 : L. paracasei CNCM I-4034 in another study was comparable in β-glucuronidase activity with L. paracasei IDCC 3401 [51].

L386-389, Pages 10 : This result was comparable that CFS of L. rhamnosus NCDC953 and L. reuteri NCDC958 inhibited the growth of Bacillus cereus, Salmonella Typhimurium, Enterococcus faecalis, and Staphylococcus aureus [54].

Reviewer 2 Report

Comments and Suggestions for Authors

The manuscript is quite interesting and well-written, addressing a topic of relevance to the food industry, especially those involved in the processing of dairy. The article's structure is appropriate and well-prepared, demonstrating high quality. I would like to suggest some minor revisions to improve it:

1 - Include approval from the animal research ethics committee. The approval is not mentioned, and the authors worked with rodents.

2 - Present the information from Tables 1, 2, and 3 as text, as they do not contribute valuable information to the manuscript, which already contains a substantial number of tables.

3 - Double-check the usage of mandatory references, such as in line 110 concerning the EFSA guidelines.

4 - Integrate a practical application of the research into the discussion. Is this microorganism already used in food production? In what ways could it be utilized?

5 - Consider dividing the discussion into additional paragraphs for improve the manuscript.

Author Response

Ms. Ref. No.: microorganisms-2771755

Title: Isolation, characterization, and safety evaluation of the novel probiotic strain Lacticaseibacillus paracasei IDCC 3401 via genomic and phenotypic approaches

Reviewers' comments:

Reviewer #2: The manuscript is quite interesting and well-written, addressing a topic of relevance to the food industry, especially those involved in the processing of dairy. The article's structure is appropriate and well-prepared, demonstrating high quality. I would like to suggest some minor revisions to improve it:

Response: We thank the reviewer for the comments made about this manuscript. We have divided and answered for each comment made by reviewer as below.

  1. Include approval from the animal research ethics committee. The approval is not mentioned, and the authors worked with rodents.

Response: We thank the reviewer for the suggestion. As suggested by the reviewer, we have added the approval number.

L144-146, Pages 3 : According to the OECD guidelines for testing chemicals [26], a single-dose acute oral toxicity test (No. TGK-2022-000580) was conducted at the Korea Testing and Research Institute (Hwasun, Korea).

  1. Present the information from Tables 1, 2, and 3 as text, as they do not contribute valuable information to the manuscript, which already contains a substantial number of tables.

Response:. We thank the reviewer for the suggestion. As suggested by the reviewer, we have changed tables to supplementary data.

  1. 3 - Double-check the usage of mandatory references, such as in line 110 concerning the EFSA guidelines.

Response: We thank the reviewer for the suggestion. As suggested by the reviewer, we have added the reference.

L111-112, Pages 3 : Determination of resistance and susceptibility against each antibiotic followed the EFSA guidelines [24].

  1. Integrate a practical application of the research into the discussion. Is this microorganism already used in food production? In what ways could it be utilized?

Response: We thank the reviewer for the suggestion. As suggested by the reviewer, we have added the practical application.

L396-401, Pages 10 : Lactobacillus paracasei is used as a microbial food culture (MFC) and probiotic [57]. It is also known to be used as a starter culture for milk fermentation, promoting flavor development in dairy products and improving functional properties in meat products [58]. In this study, the safety and characteristics of L. paracasei IDCC 3401 as a probiotic were confirmed, and it is expected that it will be possible to use it as a probiotic health functional food in the future.

  1. Consider dividing the discussion into additional paragraphs for improve the manuscript.

Response: Thanks for your suggestions. It is tricky to divide the discussion into additional paragraphs. We would like to publish at the present form.
